# Perivascular Epithelioid Cell Tumor (PEComa) of the Sigmoid Colon: Case Report and Literature Review

**DOI:** 10.3390/curroncol32060330

**Published:** 2025-06-03

**Authors:** Gintare Slice, Rokas Stulpinas, Tomas Poskus, Marius Kryzauskas

**Affiliations:** 1Faculty of Medicine, Vilnius University, 03101 Vilnius, Lithuania; 2National Center of Pathology, Affiliate of Vilnius University Hospital Santaros Klinikos, 08406 Vilnius, Lithuania; rokas.stulpinas@vpc.lt; 3Clinic of Gastroenterology, Nephrourology, and Surgery, Institute of Clinical Medicine, Faculty of Medicine, Vilnius University, 03101 Vilnius, Lithuania; tomas.poskus@santa.lt (T.P.); marius.kryzauskas@santa.lt (M.K.)

**Keywords:** perivascular epithelioid cell tumor, PEComa, colon, rectal, gastrointestinal, neoplasm, tumor, surgery, Versius

## Abstract

Perivascular epithelioid cell tumors (PEComas) are rare mesenchymal neoplasms characterized by perivascular epithelioid cell proliferation. They can occur in various organs, but colonic PEComas are exceptionally rare, showing diagnostic challenges due to their nonspecific clinical presentation and similar features to those of other colorectal tumors. We present a case of a 61-year-old female with defecation accompanied by blood clots, initially diagnosed with a suspected tumor in the sigmoid colon. Despite initial biopsy yielding non-informative material, repeat colonoscopy and imaging studies revealed a malignant tumor with multinucleated giant (osteoclast-like) cells and probable p53 mutation, most likely of mesenchymal origin. Robotic surgical resection was performed, and ultimately pathological examination refined the diagnosis as a malignant PEComa of the colon. This case demonstrates the importance of considering PEComa in the differential diagnosis of colonic tumors. Further research is needed to ascertain the clinical behavior and optimal treatment for colonic PEComas.

## 1. Introduction

Perivascular epithelioid cell tumor (PEComa) was first described in 1992 as a tumor originating from perivascular mesenchymal cells [1]. These tumors typically develop in the retroperitoneum, visceral organs, and abdominopelvic regions and commonly express markers associated with melanocytic and smooth muscle cells [2]. PEComas are most frequently diagnosed in middle-aged women, although a small number of cases have been reported in children [3,4,5,6,7,8,9,10,11]. It is important to differentiate PEComas from gastrointestinal stromal tumors, leiomyosarcomas, metastatic melanoma, chromophobe renal cell carcinoma, clear cell sarcoma, and other tumors characterized by clear cell components.

The clinical presentation of gastrointestinal PEComas is often nonspecific, with patients commonly presenting with symptoms such as abdominal pain, gastrointestinal bleeding, or obstructive symptoms. Due to the rarity of these tumors and their overlapping features with those of other mesenchymal tumors, there are no typical clinical or imaging manifestations or endoscopic characteristics to diagnose PEComas. Therefore, the diagnosis mostly relies on pathological findings [12].

PEComas characteristically show co-expression of melanocytic markers (HMB-45, Melan-A/MART1) and smooth muscle markers (SMA, desmin), with HMB-45 being the most sensitive and diagnostically useful marker. The coexpression of these markers, along with the presence of TFE3 nuclear staining in some cases and negative staining for S100, and cytokeratin and CD117 (c-kit) staining, helps distinguish PEComas from other mesenchymal tumors such as GISTs and leiomyosarcomas [2,13]. A defining feature of these tumors is the presence of somatic TSC1/TSC2 gene mutations, which result in uncontrolled mTORC1 signaling [14].

The primary treatment for PEComas and the prevention of local recurrence or distant metastasis includes the removal of the lesion [13]. The role of adjuvant therapy in preventing recurrence or metastasis is not well-established, because these tumors are characterized by high resistance to radiation and chemotherapy [14]. Furthermore, the prognosis of colonic PEComas is variable, with some tumors demonstrating indolent behavior (as shown in most cases), and others exhibiting aggressive clinical courses [15,16].

In this report, we present a case of PEComa arising in the colon, emphasizing the rarity of this entity and the importance of considering PEComa in the differential diagnosis of colonic tumors with difficult histological verification.

## 2. Methods

To conduct the search and review, the Preferred Reporting Items for Systematic Reviews and Meta-Analyses (PRISMA) guidelines were followed. A search of the PubMed database was carried out using different combinations of the following keywords: (“perivascular epithelioid cell tumor” or “PEComa”) and (“gastrointestinal tract” or “GI” or “cecum” or “colon” or “colorectal” or “sigmoid” or “rectum”). Two authors (G.S. and M.K.) independently identified relevant abstracts and then obtained the full-text articles for further review. An additional manual search of the reference lists of the included studies was performed to ensure a comprehensive search procedure (Figure 1).

Data, including study characteristics (first author name and year), general data (patient gender and age), symptoms, tumor characteristics (size, localization, metastasis), treatment, and outcomes, were extracted from eligible studies.

A case of primary colonic PEComa treated with colorectal surgery at Vilnius University Hospital Santaros Klinikos (Vilnius, Lithuania) was also included in the final table, alongside those reported in the literature, and is presented as a separate case report.

The patient provided informed consent for surgical procedure and for publication of the case report. As this was a retrospective observational study involving a literature review, institutional review board approval was not necessary.

### 2.1. Clinical Case

A 61-year-old female patient had complaints of defecation with blood clots for about 3 days. The patient underwent a colonoscopy, during which a tumor in the descending colon was found, narrowing the opening and bleeding upon contact. A biopsy was taken from it, and histology revealed a fibrinous exudate (non-informative material); therefore, a repeated colonoscopy was scheduled after a couple of weeks. The colonoscopy results showed a 4 cm tumor in the sigmoid colon, located 35 cm from the anus. It was firm and bled easily upon contact. The biopsy report revealed a malignant tumor with multinucleated giant (osteoclast-like) cells and probable p53 mutation, most likely of mesenchymal origin.

Additionally, the patient underwent computed tomography scans of the chest, abdomen, and pelvis, revealing a longer sigmoid colon forming a loop in the left inguinal area. Several small lymph nodes with a diameter of 7–8 mm were observed along the inferior mesenteric artery, along with a few slightly enlarged infrarenal and para-aortal lymph nodes up to 1 cm diameter in the abdomen. No metastases were observed in the lungs or liver.

Considering all the examinations, it was decided to proceed with surgical treatment. A robot-assisted sigmoid resection surgery was performed. A tumor of the sigmoid colon with extramural growth was found during the operation. The tumor size was 3 × 3 × 3 cm, and the tumor was mobile, non-penetrating. No signs of cancerous lymphangitis were found. The sigmoid colon was removed, and intestinal continuity was restored. The specimen taken during the operation was sent for further examination (Figure 2). No complications were observed post-surgery. The patient with a satisfactory clinical condition was discharged home six days after the surgery. There was no clinical or radiological evidence of recurrence or metastases 9 months after surgery.

### 2.2. Pathologic Examination

Gross examination revealed a 23 cm colon segment containing a 2.5 × 3 × 4 cm ulcerated, heterogeneous tumor with yellow and brown areas that bulged toward, but did not penetrate, the serosa (Figure 3). The resection margins were negative. Thirty-one lymph nodes were identified.

Microscopic examination demonstrated a tumor invading the mucosa, submucosa and muscular layer, composed of solid areas of medium-to-large epithelioid cells with irregular anisochromatic nuclei containing one to several prominent eosinophilic nucleoli. Giant cells exhibited one to several pleomorphic nuclei with marginal chromatin and macronucleoli, accompanied by abundant bright eosinophilic or vacuolated PAS+/PASD+ cytoplasm (Figure 4). The tumor showed areas of hemorrhage, with siderophages, and of focal necrosis, with ulceration of the luminal surface. Mitotic activity was 3/50 high-power fields. A perineoplastic lymphocytic infiltrate with lymphoid follicles was present at the invasion front.

The immunohistochemical findings are presented in Table 1. The final pathological diagnosis was malignant perivascular epithelioid cell tumor (PEComa) of the colon. According to the TNM staging system for visceral organ sarcomas, it was classified as follows: pT1 (4 cm tumor confined to the colon wall), N0 (0/31; all lymph nodes were negative for metastasis), LVI-0 (no lymphovascular invasion), R0 (tumor-free resection margins).

### 2.3. Results of the Literature Review

We identified 157 records through the database search and 3 additional records by screening the references. In total, 160 articles were screened based on title and abstract. After this initial screening, 49 full-text articles were reviewed, of which 10 were excluded due to the lack of patient-specific data on PEComa. Ultimately, 39 articles were included, providing information on 45 patients with primary colonic PEComa. Including our case, the total number of patients analyzed was 46 (Table 2).

## 3. Discussion

According to the 2020 World Health Organization (WHO) tumor classification, perivascular epithelioid cell tumors are defined as “members of a family of mesenchymal neoplasms composed of perivascular epithelioid cells (PECs) that express melanocytic and smooth muscle markers”. PEComas manifest in a wide range of clinical forms, from tumors with an indolent course to others resembling aggressive soft tissue sarcomas. The symptoms depend on the location and size of the tumor. PEComa’s clinical presentations include abdominal pain, melaena, rectal bleeding, obstruction, weight loss, anemia, and even lack of symptoms [2]. The gastrointestinal tract is one of its most common anatomic sites of origin, accounting for 20% to 25% of all reported cases of PEComas [14]. PEComas in general are especially rare. This often leads to a diagnostic dilemma, as this tumor type may not initially be considered in the differential diagnosis of colonic tumors.

This case highlights the diagnostic challenges associated with colonic PEComas and deciding on its treatment. The patient presented with rectal bleeding, and a tumor was found in the sigmoid colon during examination. No distant metastases were detected in this case, but the presence of enlarged lymph nodes raises concerns, as regional lymph node involvement may occur in some cases [4,10,15]. Most commonly, metastases are observed in the lungs [44], lymph nodes, and liver [2,16].

The pathological findings in this case highlight the diagnostic challenges of PEComas. The initial biopsy was non-diagnostic, underscoring the difficulties in obtaining adequate tissue sampling for a definitive diagnosis of these rare tumors. However, the persistence in diagnostic attempts, including a second colonoscopy and computed tomography scans, ultimately led to the identification of a malignant neoplasm of probable mesenchymal origin in the sigmoid colon; yet, the final PEComa diagnosis was made only after surgery, during further examination of the specimen. The tumor’s histological features—such as epithelioid cells with irregular nuclei, prominent eosinophilic nucleoli, giant cells with pleomorphic nuclei, and abundant PAS+/PASD+ cytoplasm—were consistent with PEComa. However, these features can overlap with those of other mesenchymal neoplasms, particularly gastrointestinal stromal tumors (GISTs) or leiomyosarcomas [14]. Therefore, immunohistochemical analysis is crucial for accurate diagnosis. Expression of HMB-45 is a key diagnostic marker for PEComa. While metastatic melanoma is also HMB-45-positive, it can be distinguished by its S100 protein positivity and lack of myogenic marker expression [45].

The pathologic features suggesting a potentially malignant PEComa include tumor size greater than 5 cm, infiltrative growth pattern, high nuclear grade, hypercellularity, mitotic rate greater than 1/50 HPFs, necrosis, and predominance (>50%) of atypical epithelioid cells [46]. Since it is not possible to predict whether a tumor is certainly benign or malignant, the primary treatment for the disease is often surgical resection [47].

Gastrointestinal PEComas reveal a broad spectrum of clinical outcomes. For instance, larger tumors or those with high mitotic activity tend to have a worse prognosis. This case’s tumor, with a size of 3 cm and moderate mitotic activity, indicated a moderate degree of potential aggressiveness. Malignant PEComas of the colon are extremely rare (Table 2) and reveal a broad spectrum of clinical outcomes.

Close clinical surveillance, along with imaging and colonoscopy, is essential for long-term survival due to the potential for local recurrence and distant metastasis even after a few years [7]. There is ongoing research into mTOR inhibitors as a potential therapy for this extremely rare tumor, as no effective therapy has been established [48]. According to Wagner et al., tumors within the PEComa family show a high frequency of TSC1 or TSC2 mutations [49]. Under normal circumstances, mTOR regulates cell proliferation, autophagy, and apoptosis by participating in multiple signaling pathways in the body. Together, the TSC1 (hamartin) and TSC2 (tuberin) proteins form a GTPase-activating protein complex that inactivates the small GTPase Rheb. Normally, this suppresses Rheb activity and keeps mTORC1 signaling—crucial for cell growth—under control. However, in tumor cells, abnormally activated mTOR sends signals that encourage tumor cells to grow [50,51]. Therapies targeting mTOR have certain limitations. First-generation mTOR inhibitors (rapalogs) mainly inhibit the mTORC1 complex but do not directly inhibit mTORC2, which limits their overall effectiveness. Rapalogs are metabolized by CYP3A4; so, co-administration with CYP3A4 inhibitors should be performed. Cancer cells may develop resistance to mTOR inhibitors over time by adapting or engaging alternative signaling pathways. To address these issues, combination therapies involving mTOR inhibitors alongside other anticancer agents are being investigated to improve treatment efficacy and overcome resistance mechanisms. Combination therapies may prevent the compensatory activation of alternative signaling pathways that arise after single-agent mTOR inhibition [52].

Despite its rarity, awareness of PEComa as a potential differential diagnosis of colonic tumors is crucial. However, due to the rarity of the disease, there are still significant challenges in performing a therapeutic trial.

## 4. Conclusions

Perivascular epithelioid cell tumors of the colon represent a very rare entity that poses significant diagnostic challenges. Their accurate diagnosis requires careful integration of morphologic findings, immunohistochemical studies, and clinical parameters. While surgical resection remains the primary treatment, the variable biological behavior of these tumors emphasizes the importance of thorough pathologic assessment of risk factors and close clinical follow-up. This case highlights the value of a comprehensive pathologic evaluation in establishing the correct diagnosis and guiding the clinical management of rare mesenchymal tumors.

## Figures and Tables

**Figure 1 curroncol-32-00330-f001:**
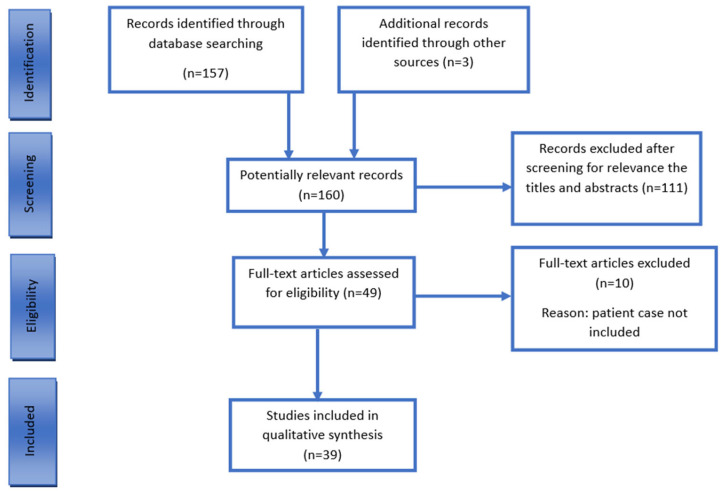
PRISMA flow chart of literature search. PRISMA = Preferred Reporting Items for Systematic Reviews and Meta-Analyses.

**Figure 2 curroncol-32-00330-f002:**
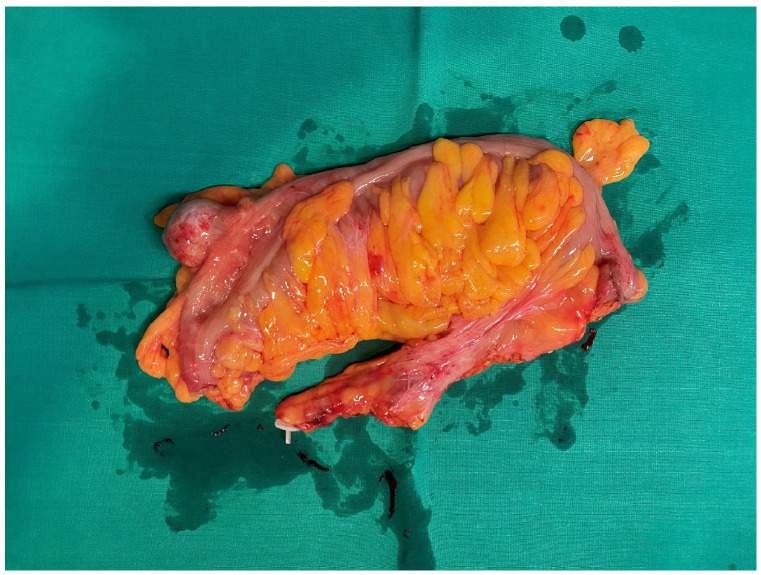
Removed part of the colon with the tumor.

**Figure 3 curroncol-32-00330-f003:**
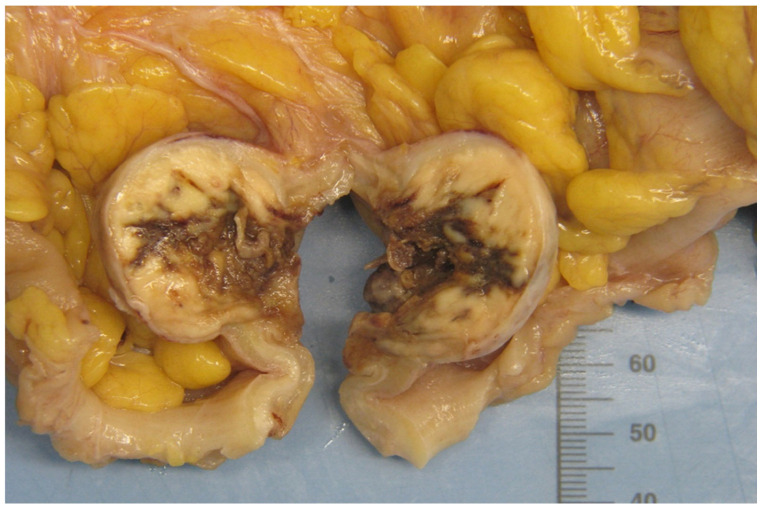
Gross pathologic specimen of sigmoid colon showing a heterogeneous tumor with yellow-brown cut surface.

**Figure 4 curroncol-32-00330-f004:**
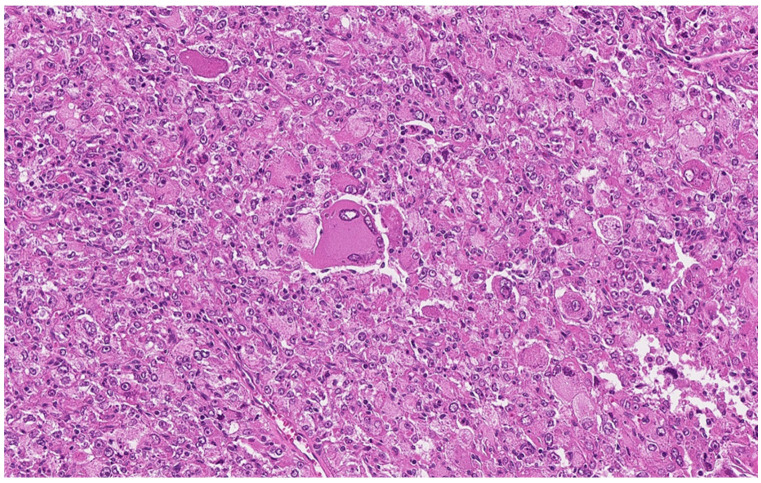
Histopathologic features of the colonic PEComa demonstrating sheets of epithelioid cells with abundant eosinophilic cytoplasm, nuclear pleomorphism, and prominent macronucleoli. Note the presence of a multinucleated giant cell in the center (H&E stain, ×200 magnification).

**Table 1 curroncol-32-00330-t001:** Immunohistochemical findings of the primary colon PEComa case.

Markers	Staining Intensity	Positive Cells (%)
Cathepsin K	++	100%
CD68	+/++	5%
MiTF	+++	10%
HMB45	++	2%
Vimentin	+/+++	30%
TFE3	+/++	30%
Smooth muscle actin	+	5%
CyclinD1	+/++	40%
Ki-67	+	10%
Desmin	++/+++	60%
MelanA/HMB	+	<1%
Pancytokeratin (AE1/AE3)	−	−
Caldesmon	−	−
Myogenin	−	−
MyoD1	−	−
DOG1	−	−
CD117	−	−
S100	−	−
SOX-11	−	−
ALK1	−	−
MUM1	−	−

**Table 2 curroncol-32-00330-t002:** Review of case reports of colorectal perivascular epithelioid cell tumor.

Authors	Year	Gender	Age	Symptoms	Tumor Size (CT or Colonoscopy)	Localization	Metastasis	Initial Treatment	Outcomes
Prasad et al. [17]	2000	Female	22	None (incidental)	3 cm	Cecum	No	Surgery	NED at 6 mo
Tazelaar et al. [18]	2001	Female	9	Tumor prolapse per anus	3 cm	Rectum	No	Surgery	NED at 14 mo
Tazelaar et al. [18]	2001	Female	40	Profuse rectal bleeding	3 cm	Rectum	No	Surgery	NED at 6 mo.
Birkhaeuser et al. [19]	2004	Female	35	Occult blood in the feces	3–4 cm	Cecum	No	Surgery	NED at 9 mo
Genevay et al. [20]	2004	Female	36	Anemia and rectorrhagia	3.5 cm	Cecum	No	Surgery	NA
Genevay et al. [20]	2004	Female	35	Pararectal mass	-	Rectum	No	Surgery	NA
Evert et al. [21]	2005	Female	56	Rectal obstruction, gluteal pain, recurrent diarrhea, weight loss	8 × 5 cm	Rectum	Lung metastasis	Surgery	NA
Yamamoto et al. [22]	2006	Female	43	Abdominal pain	8 cm	Descending colon	No	Surgery	Peritoneal dissemination of tumor at 20 mo DOD at 36 mo
Baek et al. [23]	2007	Female	16	Rectal bleeding	2.5 × 2.0 cm	Proximal transverse colon	No	Endoscopic resection	NED at 24 mo
Pisharody et al. [24]	2008	Male	11	Rectal bleeding	3 cm	Sigmoid colon	Lymph node metastasis	Surgery	NED at 5 mo
Righi et al. [25]	2008	Male	11	Rectal bleeding	3.5 cm	Descending/sigmoid colon	No	Surgery	NA
Qu et al. [26]	2009	Female	43	-	2 cm	Cecum	No	Surgery	NED at 25 mo
Ryan et al. [10]	2009	Female	15	Rectal bleeding	4 cm	Rectum	Lymph node metastasis	Surgery and adjuvant chemotherapy	NED
Tanaka et al. [11]	2009	Female	14	Physical examination	4 cm	Sigmoid colon	No	Surgery	NED at 5 mo
Gross et al. [7]	2010	Male	5.5	Abdominal pain and fever	5 cm	Ascending colon	No	Surgery	NED at 2 yr
Freeman et al. [8]	2010	Female	17	Rectal bleeding	4–6 cm	Sigmoid colon	No	Surgery	NED at 15 yr
Park et al. [9]	2010	Male	7	Abdominal pain	3.9 cm	Ascending colon	No	Surgery	NED at 26 mo
Shi et al. [27]	2010	Female	38	-	6.0 cm	Ascending colon	No	Surgery	NED at 8 mo
Shi et al. [27]	2010	Male	42	-	4.5 cm	Sigmoid colon	No	Surgery	NED at 15 mo
Shi et al. [27]	2010	Male	36	-	4.8 cm	Descending colon	No	Surgery	NED at 32 mo
Shi et al. [27]	2010	Female	45	-	3.5 cm	Ascending colon	No	Surgery	NED at 36 mo
Maran-Gonzalez et al. [6]	2011	Female	11	Prolapsed mass	2 cm	Rectum	No	Endoscopic resection;	NA
Lee et al. [28]	2012	Female	62	Bleeding	5 cm	Sigmoid	No	Surgery	NED at 16 mo
Lee et al. [28]	2012	Female	62	Abdominal pain and melena	5.5 × 4.2 × 2.5 cm	Sigmoid colon	No	Surgery	NED at 4 mo
Scheppach et al. [29]	2013	Male	23	Abdominal pain and melaena	5.5 cm	Cecum	Lymph node and liver metastasis	Surgery and adjuvant chemotherapy	DOD
Im et al. [5]	2013	Male	17	Hematochezia	3 cm	Rectum	No	Endoscopic resection	NED at 10 mo
Korytnaya et al. [30]	2014	Female	47	Amenorrhea, galactorrhea, abdominal pain	17 × 13 × 8 cm	Distal stomach, proximal duodenum, and right colon	No	Surgery	NED
Kanazawa et al. [31]	2014	Female	55	None (physical examination)	3 cm	Rectum	No	Endoscopic resection	NED at 15 mo
Balta et al. [32]	2015	Male	49	Rectum bleeding	-	-	-	-	-
Iwamoto et al. [33]	2016	Female	42	None ( annual medical checkup)	5 cm	Descending colon	No	Surgery	NED at 5 mo.
Cheng et al. [34]	2016	Male	40	Dyschezia	6 × 7 cm	Sigmoid colon	No	Surgery	Recurrence and pancreatic metastasis at 27 mo
Batereau et al. [35]	2016	Female	26	Localized abdominal pain	-	Transverse colon	No	Surgery	AWD, multiple relapses
Kolin et al. [36]	2017	Female	18	Rectal bleeding	1.8 cm	Rectum	-	Surgery	NA
Neuhaus et al. [37]	2018	Female	36	Abdominal and suprapubic pain	10 cm	Transverse colon	No	Surgery	NA
Lin et al. [38]	2018	Male	28	Abdominal pain and intermittent melena	8.9 × 7.2 cm	Cecum	No	Surgery	Liver metastasis at 49 mo NED at 28 mo. after the second operation
Iwa et al. [39]	2019	Male	69	None ( annual medical checkup)	4.1 × 4.0 × 3.2 cm	Cecum	No	Surgery	NA
Uhlenhopp et al. [16]	2020	Male	64	Left lower quadrant pain and nausea with vomiting	12.6 ×10 × 9 cm	Sigmoid colon	Liver metastasis	Surgery	DOD
Bennett et al. [40]	2020	Female	67	No gastrointestinal symptoms	0.8 cm	Ascending colon	No	Endoscopic dissection	NED
Fuse et al. [41]	2021	Female	53	-	-	Colon	-	Surgery	Lung metastasis at 8 yr
Cheng et al. [4]	2021	Female	17	Painless hematochezia	3.5 × 3.1 × 2.8 cm	Rectosigmoid Colon	Lymph node metastasis	Surgery and adjuvant chemotherapy	NED at 24 mo
Razak et al. [42]	2022	Female	69	None (annual medical checkup)	-	Cecum	No	Surgery	NED at 48 mo
Kou et al. [3]	2022	Female	12	Abdominal pain	5.0 × 4.5 × 3.0 cm	Transverse colon	No	Surgery	NED at 4 mo
Chen et al. [43]	2023	Female	55	Abdominal pain, weight loss	12.0 × 6.7 cm	Ascending Colon	No	Surgery	NA
Yan et al. [12]	2023	Female	47	-	1.5 × 2.0 cm	Sigmoid colon	No	Endoscopic resection	NED at 3 mo
Sugimura et al. [44]	2024	Male	64	None (follow-up colonoscopy)	0.4 cm	Transverse colon	No	Polypectomy	NED
Current report	2024	Female	61	Defecation with blood clots	3 × 3 × 3 cm	Sigmoid colon	No	Surgery	NED at 9 mo

DOD: died of disease; NA: not available; NED: no evidence of disease; AWD: alive with disease.

## Data Availability

The original contributions presented in this study are included in the article. Further inquiries can be directed to the corresponding author.

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
