# Peer review of "Perivascular Epithelioid Cell Tumor (PEComa) of the Sigmoid Colon: Case Report and Literature Review"

_curroncol, 2025, doi:10.3390/curroncol32060330_

Round 1
Reviewer 1 Report
Comments and Suggestions for Authors
Perivascular Epithelioid Cell Tumor (PEComa) of the Sigmoid Colon: Case Report and Literature
This case report describes a very rare tumor based on extensive literature data. Diagnosis is very difficult because the clinical features are very similar to those of gastrointestinal tumors, leiomyosarcomas, or other clear cell tumors.
Without the specific mesenchymal markers listed in the manuscript, the clinician cannot differentiate this tumor from other mesenchymal tumors.
The number of undetected tumors is almost impossible to determine.
Despite the rarity of this tumor, the authors have compiled comprehensive and insightful data on this topic through extensive literature research. For this reason alone, and due to the very concise presentation of the problem and the clinical case, this study offers surgical clinicians a very useful overview and valuable expertise.
I recommend publishing this work in its current form.
Author Response
Thank you for your insight.
Reviewer 2 Report
Comments and Suggestions for Authors
The comments are in the attached word file.

Author Response
My response to Reviewer 2's comments:- Comment: Expand on mTOR pathway biology - briefly explain how TSC1/2 mutations lead to mTORC1 activation and tumor growth, should be further explained Response: I have added some additional information to the paper, which is highlighted in the revised manuscript text
- Comment: For a more rigorous discussion of mTOR-targeted therapy, discuss limitations of therapy such as variability and durability of responses, and discuss combination therapies Response: I have added some additional information to the paper, which is highlighted in the revised manuscript text
- Comment: Adding a summary chart might be helpful in better communicating pathways and therapies Response: We chose not to add additional summary charts
- Comment: Improve clarity by doing less switching between “tumor,” “neoplasm,” and PEComa without specific contextual reasons to do so Response: We changed a couple of "neoplasm" to " tumor" but we believe that in some cases the term "neoplasm" is more appropriate than "tumor", while in other instances the term "PEComa" is more suitable for the context.
- Comment: Consider summarizing key findings visually through the use of chart Response:We chose not to add additional charts
Reviewer 3 Report
Comments and Suggestions for Authors
Congratulations on an interesting case, solved through minimally invasive, robotic resection.
The case study is well-described, and the methodology, including the PRISMA flow diagram, is presented nicely.
However, I kindly request clarification from the authors regarding the statement: "No signs of cancerous lymphangitis were found."
Do you mean that no intraoperative signs of lymph node involvement were found, or are you referring to the absence of inflammation in the lymphatic vessels?
Author Response
Thank you for your insight. To answer the question, we were referring to the absence of inflammation in the lymphatic vessels.